# A Spectroscopic and Molecular Dynamics Study on the Aggregation Properties of a Lipopeptide Analogue of Liraglutide, a Therapeutic Peptide against Diabetes Type 2

**DOI:** 10.3390/molecules28227536

**Published:** 2023-11-11

**Authors:** Micaela Giannetti, Antonio Palleschi, Beatrice Ricciardi, Mariano Venanzi

**Affiliations:** Department of Chemical Science and Technologies, University of Rome Tor Vergata, Via della Ricerca Scientifica 1, 00133 Rome, Italy; micaela.giannetti@uniroma2.it (M.G.); antonio.palleschi@uniroma2.it (A.P.); beatrice.ricciardi@uniroma2.it (B.R.)

**Keywords:** liraglutide analogue, molecular dynamics simulations, peptide aggregation, peptide nanostructures, therapeutic peptides

## Abstract

The pharmacokinetics of peptide drugs are strongly affected by their aggregation properties and the morphology of the nanostructures they form in their native state as well as in their therapeutic formulation. In this contribution, we analyze the aggregation properties of a Liraglutide analogue (LG18), a leading drug against diabetes type 2. LG18 is a lipopeptide characterized by the functionalization of a lysine residue (K26) with an 18C lipid chain. To this end, spectroscopic experiments, dynamic light scattering measurements, and molecular dynamics simulations were carried out, following the evolution of the aggregation process from the small LG18 clusters formed at sub-micromolar concentrations to the mesoscopic aggregates formed by aged micromolar solutions. The critical aggregation concentration of LG18 in water (pH = 8) was found to amount to 4.3 μM, as assessed by the pyrene fluorescence assay. MD simulations showed that the LG18 nanostructures are formed by tetramer building blocks that, at longer times, self-assemble to form micrometric supramolecular architectures.

## 1. Introduction

Peptides exert their biological activity as hormones, neurotransmitters, growth factors, molecular triggers, and antibiotics. These manifold functions make peptides versatile lead compounds for drug design and therapy, especially in view of their high biocompatibility, target selectivity, and low toxicity [1,2,3]. Unfortunately, peptide drugs generally present low bioavailability, low permeability and stability in the membrane, and fast pharmacokinetics in serum due to rapid enzymatic degradation [4]. Apart from that, at the present time, intravenous or subcutaneous administrations are the most efficient ways of peptide drug vehiculation, despite intense research efforts devoted to developing less invasive methods [5].

A strategy to improve the efficiency of therapeutic peptides, recently adopted with some success, is their functionalization with lipid chains, creating a new class of compounds denoted as lipopeptides (LP) [6,7]. Lipidation has been shown to prevent enzymatic degradation and reduce renal clearance of peptide drugs [8]. The LP amphiphilic properties can be suitably tuned by varying the length and composition of the lipid chain and balancing the number of non-polar/polar amino acids in the peptide sequence [9]. While the peptide moiety is responsible for the drug targeting capacity and selectivity, the lipid fraction promotes the formation of LP aggregates at very low concentrations and favors the interaction of LP drugs with the membrane phase [10].

LP drugs have shown unique pharmaceutical properties in terms of high metabolic stability, long circulation times, and high affinity for serum proteins [11]. However, although peptide lipidation is currently well developed, the mechanism by which LP drugs exert their therapeutic activity is still poorly understood [12]. Two possible explanations have been advanced about the unique LP pharmacokinetic properties: (i) the formation of LP micelles resilient to enzymatic degradation and lung elimination, and (ii) the protective binding of LP to serum proteins. Therefore, the study of LP aggregation and the formation and morphological characterization of LP nanostructures are important for the comprehension of the degradation mechanisms determining LP pharmacokinetics.

Liraglutide (LG), a lipidated glucagon-like peptide 1 derivative, more than a decade ago became a once daily leading drug in the therapy of diabetes and obesity [13]. However, despite its prominent role in the market, LG pharmacokinetics is still under scrutiny [14]. Former studies have shown that LG complexation with human serum albumin (HSA) is one of the factors affecting LG circulation time and catabolism [15]. Moreover, it has been shown that the relatively large size of HSA protects LG against proteolytic degradation or rapid renal filtration, resulting in a long half-life in humans (11–15 h) where it is vehiculated by subcutaneous administration [16]. It has also been claimed that LP aggregation could contribute significantly to its prolonged blood circulation [17]. In this context, it has been shown that the dimensions of LG aggregates are strongly dependent on the pH, with LG aggregates being formed by eight LP units at pH 8 [18] and twelve LP units at pH 6.4 [19]. Innovative procedures have also been applied to the synthesis of LG using total chemical routes [20].

In this contribution, we report on the aggregation properties of an LG analogue, denoted in the following as LG18, by spectroscopy methods and molecular dynamics (MD) simulations. LG18 comprises a 32 amino acid long peptide chain functionalized at the side chain amino group of K26 by an 18C lipid chain (Figure 1). As LG18 features an isoelectric point at around peptide unit four [18], and the present studies were carried out at pH 8.1, LG18 should be considered negatively charged.

Interestingly, NMR spectroscopy experiments on LG18 have found an almost constant hydrodynamic radius (R_h_) of 28 Å in the concentration range between 50 and 250 μM, a value compatible with LG18 aggregates formed by eight peptide units [21]. The observation that a similar R_h_ was also found in the case of 20 μM LG18 solutions led the authors to hypothesize a critical aggregation concentration (*cac*) of about 1 μM [21].

Fluorescence and Dynamic Light Scattering (DLS) techniques can be routinely applied in the 0.1–10 μM concentration range, complementing the NMR region and allowing for the determination of the hydrodynamic radius of LG18 oligomers at micro- and sub-micromolar concentrations. The LG18 Y19 and W31 residues are used as intrinsic probes for fluorescence studies. MD simulations were also carried out to analyze the first steps of the LG18 aggregation process, characterizing the formation of the small oligomeric clusters leading, on a longer time scale, to mesoscopic peptide aggregates.

## 2. Results

### 2.1. Spectroscopic Characterization of LG18

The UV-vis absorption spectrum of LG18 (20 μM, PBS, pH = 8.1) between 200 and 350 nm is characterized by two absorption maxima at 211 nm and 280 nm, associated with the π–π* (^1^L_a_ and ^1^L_b_) transitions of the Y19 and W31 aromatic side chains (supporting materials, Appendix A). The n–π* amide transition associated with the amide bond of the peptide backbone also weakly contributes to the absorption in the far UV region.

Circular Dichroism (CD) experiments showed that LG18 in PBS aqueous solution at micromolar (2–10 μM) concentrations and pH 8.1 attains a helical conformation (Figure 2). Interestingly, the LG18 helical secondary structure was found to be stably conserved two weeks after the preparation of the LG18 solution.

Steady-state fluorescence spectra of LG18 at λ_ex_ = 280 and 295 nm (λ_em_ = 300–500 nm) are reported in Figure 3A. Both the fluorescence spectra show an emission maximum at λ_em_ = 350 nm and closely overlapping spectral shapes, independently of the excitation wavelength. This finding indicates an almost complete Y19*→W31* excitation energy transfer, suggesting that LG18 attains a rather compact structure. The 2D excitation/emission spectrum reported in Figure 3B confirmed that the fluorescence emission of LG18 is due to a single emitting species, i.e., the W31 moiety, characterized by an emission centered at λ_em_ = 350 nm.

Time-resolved fluorescence emission measurements on LG18 (21 μM, pH = 8.1, λ_ex_ = 295 nm, λ_em_ = 350 nm) showed that the W31 time decay can be accounted for by three lifetime components, i.e., τ_1_ = 1.22 ns, τ_2_ = 3.66 ns, and τ_3_ = 7.40 ns, with fractional weights equal to α_1_ = 0.45, α_2_ = 0.41, and α_3_ = 0.14, respectively. This finding suggests that the W31 indole side chain experiences different local environments because of the conformational heterogeneity of LG18. From these data, an average lifetime, i.e., <τ> = 3.0 ns, can be assigned to the LG18 fluorescence emission time decay.

The steady-state fluorescence anisotropy coefficient was measured under the same experimental conditions, obtaining *r* = 0.110 (±0.001). Assuming a globular shape of LG18, the fluorescence anisotropy coefficient can be related to the overall rotational correlation time, *σ*, through the fluorescence lifetime τ (spherical rotor approximation):$$\frac{r_{0}}{r}=1+\frac{\tau }{\sigma },$$
where *r*_0_ is the limit fluorescence anisotropy coefficient, i.e., the anisotropy coefficient = 0. Assuming *r*_0_ = 0.260 [22] and τ = 3 ns, we obtained *σ* = 2.2 ns.

From the rotational correlation time, the hydrodynamic volume (*V_h_*) of LG18, considered a spherical rotor, can be obtained:$$\sigma =\frac{\eta V_{h}}{RT},$$
where *η* is the viscosity coefficient of the solvent (0.89 cP at T = 298 K). From these data, the hydrodynamic volume of LG18 under the applied experimental conditions is estimated to be 10.2 nm^3^, while its hydrodynamic radius, R_h_, amounts to 13.4 (±0.1) Å.

### 2.2. LG18 Aggregation Studies

#### 2.2.1. Spectroscopic Studies

The first evidence of LG18 aggregation came from the analysis of the UV-vis absorption spectrum of a 21 μM LG18 solution at pH = 8.1 two weeks after its preparation. The absorption spectrum appeared broader and strongly affected by diffuse light (Figure 4A) when compared with the spectrum measured for a freshly prepared LG18 solution at the same concentration. The structureless transition in the 250–300 nm region strongly suggests that LG18 aggregation could be favored by intra- and/or intermolecular π–π stacking interactions between the aromatic side chains of Y19 and W31.

This idea is confirmed by the fluorescence emission spectra of LG18, measured under the same experimental conditions (Figure 4B). The fluorescence emission measured for a two week old solution is notably quenched, broadened, and sensibly shifted to shorter wavelengths with respect to the emission spectrum measured on an LG18 freshly prepared solution. This finding strengthens the hypothesis of the occurrence of intra/intermolecular stacking interactions involving the Y19 and W31 aromatic moieties, giving rise to the formation of sandwiched dimers (H-type aggregates).

Interestingly, the excitation spectra (λ_em_ = 350 nm) of both freshly prepared and aged LG18 solutions, reported as SM (Appendix A), are typical of the W fluorophore emission as a monomer species, suggesting the formation of non-fluorescent aggregates (H-type) in the UV region.

Strong evidence of aggregation has been provided by the emission spectra measured at λ_ex_ = 350 nm (λ_em_ = 310–500 nm) of one month aged LG18 solutions at micromolar concentrations (Figure 5). Under the applied experimental conditions (10 μM), the emission spectrum of LG18 is dominated by a weak, but clearly discernible, band that peaked at 440 nm. This emission is typical of peptide amyloids, and its occurrence has been ascribed to the formation of large peptide aggregates characterized by an extended network of intermolecular hydrogen bonds [23,24]. The time-dependent growth of this peculiar emission was also studied in detail in the case of Semaglutide, a peptide analogue of Liraglutide currently employed against diabetes 2 and characterized by a two lipid chain functionalization [22].

In Figure 5A, the sharp Raman band at 395 nm (ascribable to water molecules) was shown with the aim of highlighting the weak intensity of the emission band, which peaked at 440 nm. Interestingly, the excitation spectrum of LG18 measured at λ_em_ = 450 nm (Figure 5B) revealed that this peculiar emission is generated by a low-energy state characterized by an absorption band peaked at 350 nm.

Steady-state fluorescence anisotropy experiments (λ_ex_ = 295 nm and λ_em_ = 350 nm) were also carried out for LG18 solutions at micromolar concentrations (Table 1). The results of time-resolved fluorescence measurements on the same solutions are reported as SM (Appendix A). From the anisotropy coefficients and the fluorescence time decay data and applying the spherical rotor approximation, it is possible to obtain an estimate of the hydrodynamic radius of LG18 at the different concentrations investigated (Table 1). It can be seen that the hydrodynamic radius slightly increases with the LG18 concentration, suggesting the continuous growth of small LG18 oligomers.

Diffuse light scattering experiments showed that LG18 freshly prepared solutions at concentrations ranging from 0.47 μM up to 2.8 μM only form nanometric aggregates at the limit of the experimental resolution (Appendix A). The same experiment was also carried out for a 16 μM one month aged solution of LG18. In this case, the predominant contribution to the scattered light originated from LG18 aggregates, whose diameter distribution peaked at 0.9 μm (Appendix A). When the investigated solution was filtered by a 0.2 μm filter, the larger component of the size distribution was eliminated, and a single nanometric distribution, distinguishable from the signal of the buffer solution only for a higher scattered light intensity, was detected.

To obtain a more detailed description of the aggregate size, DLS experiments were carried out on 2 and 10 μM LG18 aged (one month) solutions (PBS, pH 8.1). We found that at the lower concentration, LG18 formed aggregates featuring an average diameter of about 136 (±48) nm, while at 10 μM, the average size of LG18 aggregates grew to about 313 (±26) nm (Figure 6). Of note, while the LG18 size distributions recovered by DLS experiments at 2 μM (three replicas) are rather broad and dispersed, the size of the three-replicas distributions obtained at 10 μM appear narrower and more homogeneously dispersed, suggesting the formation of stable aggregates.

Zero potential measurements carried out on LG18 provided a value of −34 (±2) mV and −38 (±2) mV for the two concentrations investigated, respectively, confirming the formation of stable lipopeptide aggregates, characterized by a negative potential surface, as expected under the applied experimental conditions (pH 8.1).

#### 2.2.2. Determination of the LG18 Critical Aggregation Concentration (*cac*)

The pyrene assay is a fluorescence method based on the measurement of the ratio of the first (I_1_, λ_1_ = 372 nm) and third (I_3_, λ_3_ = 387 nm) vibronic components of the pyrene emission band, comprised between 360 and 420 nm. It has been shown that this ratio is highly sensitive to the polarity of the pyrene environment, so that the interaction between the pyrene fluorophore and the hydrophobic environment of the peptide aggregate can be monitored, allowing for determining the peptide *cac* [25].

The fluorescence emission spectra of a 1 μM pyrene aqueous solution for different LG18 concentrations (from 0.2 to 21 μM) were therefore recorded (Appendix A), and the I_1_/I_3_ fluorescence intensities ratio was rightly determined (Appendix A). Concerning the intensity of the pyrene emission, the results reported in Appendix A clearly show that the pyrene fluorescence is steadily quenched by increasing the LG18 concentration, most likely caused by pyrene/W and/or pyrene/Y stacking interactions.

Interestingly, reporting the I_1_/I_3_ fluorescence intensities ratio as a function of the LG18 concentration in a semi-log plot, clearly shows a discontinuity of this parameter for a critical LG18 concentration (Figure 7). At the lower LG18 concentrations, the I_1_/I_3_ ratio is almost constant, while it steeply decreases with increasing LG18 concentrations after a critical concentration value. We ascribe this concentration-dependent behavior to the formation of LG18 extended aggregates, able to embed the pyrene fluorophore in a hydrophobic environment [26]. Consequently, we obtained an estimate of the *cac* parameter at the inflection point of the (I_1_/I_3_) vs. log [LG18] curve reported in Figure 7. From these data, a *cac* of about 4.3 μM can be estimated for LG18 under the applied experimental conditions.

#### 2.2.3. Molecular Dynamics Simulations

In fair agreement with CD results, MD simulations show that, in its monomeric form, LG18 attains predominantly an α-helical conformation due to the appropriate distribution of charged and hydrophobic residues (Figure 8A). It should be noted that the LG18 α-helix is stabilized by the positioning of the charged R and E side chains on the opposite sides of the helix surface. Interestingly, in this conformation, the E9 and H7 side chains are hydrogen bonded. Furthermore, F28 and W31, both placed in the LG18 hydrophobic region extending from F28 to V33, are paired by π–π stacking interaction, representing a suitable site for pyrene intercalation. Of note, the long hydrophobic arm of K26m wraps around the peptide helix, held in this position by the formation of a salt-bridge link between the K26m side-chain carboxylate group and the positively charged R34.

As a consequence of these multiple stabilizing interactions, the α-helical conformation is very stable during the simulation time, as demonstrated by the low root mean square fluctuations (rmsf) of the C^α^-atoms of the residues situated in the central segment of the peptide chain (Appendix A). As expected, the mobility of the N- and C-termini is enhanced by the presence of the four G10, G11, G35, and G37 residues.

MD simulations performed on two LG18 units demonstrated that they could form stable dimers characterized by an antiparallel arrangement of the two peptide main chains (Figure 8B). In this case, the α-helical conformation is very stable for both peptide chains, and the formation of the adduct is mainly driven by the hydrophobic effect. Interestingly, the LG18 dimer shows an inner hydrophobic region formed by the F12, V16, Y19, and L20 residues of the first peptide and the A24, A25, F28, I29, A30, W31, L32, and V33 of the second peptide unit. The two K26m lipid arms also contribute to stabilizing the hydrophobic cavity of the peptide dimer.

In the case of MD simulations carried out with four LG18 units, a tetramer aggregate characterized by a high content of α-helical structures was obtained. The four helices form a very compact structure, stabilized by the antiparallel arrangement of the peptide helical chains (Figure 9).

Also, in the case of the tetramer, the K26m side chains play an important role in the formation of a very compact aggregate, gluing the hydrophobic regions of the four helices in an inner lipid-like domain. Overall, the tetramer shows a typical protein-like structure characterized by a hydrophobic core region and an outer surface of charged residues exposed to the solvent. The four α-helical structures are quite stable during the probed simulation time, as demonstrated by the low root mean square fluctuations (rmsf) of the C^α^ atoms (Appendix A), indicating that all the LG18 units showed reduced mobility, especially in the central region of the peptide chains where it is generally less than 1 Å.

These results strongly suggest that LG18 tetramers can be considered the building blocks of larger aggregates. However, it must be taken into account that the negatively charged side chains exposed to water may represent a limiting factor that may hinder the formation of larger aggregates. This idea is strengthened by MD simulations carried out with eight and sixteen LG18 units, where supramolecular aggregates formed by LG18 tetramers could be occasionally observed (Appendix A). These supramolecular architectures are stabilized by electrostatic interactions established among the charged residues located on the heads and tails of the peptides forming the tetramer building blocks. In Table 2, we reported the geometrical parameters, i.e., gyration radius (R_g_) and volume (V_g_), as well as the hydrodynamic radius (R_h_) and volume (V_h_) of LG18 structures from the monomer to the hexadecamer species, as determined by MD simulations. It should be noted that the R_g_ and R_h_ values of such relatively compact structures (Table 2) are comparable with the Förster Energy Transfer critical distance for the Y(donor)–W(acceptor) pair, i.e., R_0_ = 15 Å, in agreement with the observed fluorescence emission spectral features (Figure 3).

## 3. Materials and Methods

### 3.1. Materials

Synthesis and characterization of LG18 were already reported elsewhere [21]. Chemicals have been purchased from Merck Italia (Milan, Italy) and Carlo Erba (Cornaredo, Italy) and used as such without further purification.

### 3.2. Methods

*UV-vis absorption.* UV-vis absorption spectra have been carried out with a Cary100 SCAN (Agilent Technologies, Santa Clara, CA, USA) spectrophotometer using a band width of 1 nm and quartz cuvettes (Hellma Italia, Milan, Italy) with an optical length of 1 cm. Unless otherwise stated, all the spectroscopic experiments were carried out on phosphate buffered aqueous solutions (PBS) at a pH = 8.1.

*Circular Dichroism (CD).* CD experiments were carried out on 2 μM and 10 μM LG18 PBS aqueous solutions with a JASCO J800 (Jasco Europe Srl., Cremella, Italy) spectropolarimeter using quartz cells with 0.1 and 1 cm optical paths.

*Fluorescence.* Steady-state fluorescence experiments were performed on a Fluoromax-4 (Horiba, Kyoto, Japan) spectrofluorometer at 25 °C using quartz cuvettes with 1 cm optical length. Fluorescence emission and excitation spectra were carried out at different excitation (λ_ex_ = 280, 295, and 350 nm) and emission (λ_em_ = 350 and 420 nm) wavelengths, respectively, using excitation/emission (e/e) slit widths of 3 nm. Steady-state anisotropy measurements were carried out on the same spectrofluorometer equipped with automatically controlled Glan–Thomson polarizers. The fluorescence anisotropy coefficient *r* was determined at λ_ex_ = 295 and λ_em_ = 350 nm through the equation:$$r=\frac{I_{vv}-I_{vh}}{I_{vv}+2I_{vh}},$$
where *I_vv_* and *I_vh_* are the fluorescence intensity measured with the e/e polarizers oriented in the vertical/vertical and vertical/horizontal positions, respectively.

Time-resolved fluorescence experiments were carried out by an EAI Spec-ps (Edinburgh Analytical Instrument, Edinburgh, UK) using quartz cuvettes with a 1 cm optical length. Excitation at λ_ex_ = 298 nm was achieved by a diode laser with a 1 ns pulse width, while the emission was set at λ_em_ = 350 nm. Experimental parameters: 1 nm emission slit width; 50 ns time window; 1024 channels; 320 nm cut-off filter; and stop: 10^4^ counts at maximum.

For the *cac* determination, a 1 μM pyrene aqueous solution was titrated by adding aliquots of LG18 solution in the 0.2–21 μM concentration range (pyrene assay). The ratio of the intensities of the first (S_1_^v=0^→S_0_^v=0^, λ_em_ = 372 nm) and third (S_1_^v=0^→S_0_^v=1^, λ_em_ = 387 nm) vibronic components of the pyrene emission band was reported as a function of the LG18 concentration in a semi-log graph. The *cac* was therefore determined at the intersection of the two linear regions associated with the polar/apolar environments experienced by the pyrene fluorophore.

*Light Scattering.* The light scattering intensity of freshly prepared and aged LG18 micromolar aqueous solutions (pH = 8.1) was measured by an LB-550 HORIBA (Horiba, Kyoto, Japan) particle size analyzer.

*Dynamic light scattering (DLS).* DLS experiments and zeta potential measurements on 2 μM and 10 μM LG18 aqueous solutions (pH = 8.1) were carried out using a Zetasizer Nano ZS (Malvern Instruments, Malvern, UK) at room temperature using disposable cuvettes. The DLS instrument was equipped with a 5 mW HeNe laser, and the scattered light was collected at an angle of 173°.

*Molecular Dynamics (MD).* MD simulations were performed using the Charmm27 force field implemented in the GROMACS version 5.1.2 software package. The number of LG18 and H_2_O molecules and the size of the cubic simulation box of each MD simulation are reported in Table 3. The protic side chain groups of basic and acidic residues of LG18 were charged accordingly to the experimental pH. Due to the fact that at pH = 8.1, LG18 features a net negative charge (−3), three Na^+^ counterions per molecule were added to neutralize the simulated systems.

The force field parameters used for the K26 residue functionalized by the lipid chain (K26m) were determined by analogy with similar groups. The carboxylate group of the K26m side chain was set as charged, according to the experimental pH.

The *spce* (single point charge extended) water model was used for all the MD simulations performed. For all the investigated systems, two 100 ns simulations were carried out, applying periodic boundary conditions in the XYZ direction. A cutoff of 1.4 nm was applied for both Coulombic and van der Waals interactions using the potential-shift-Verlet modifier. NPT simulations were carried out in the weak coupling regime using temperature coupling velocity rescaling with a time constant of 0.6 ps and a reference temperature of 300 K. The Berendsen algorithm under isotropic conditions was applied for the pressure coupling using a time constant of 1 ps. All bonds were constrained using the LINCS algorithm, except during equilibration (a time step of 0.5 fs). The compressibility was set equal to 4.5 × 10^−5^ bar^−1^.

For each system, the radius of gyration was calculated through the *gyrate* GROMACS tool. The hydrodynamic radius was calculated using the equation:$$R_{h}={\left\langle {r_{ij}}^{-1}\right\rangle }^{-1},$$
where *i* and *j* are indexes running on the C atoms of the systems. Assuming LG18 as a rigid spherical rotor, the hydrodynamic volume was calculated through the equation:(1)$$V_{h}=\frac{4}{3}\pi {rR}_{h}^{3}$$

## 4. Conclusions

The aggregation properties of a lipopeptide analogue (LG18) of Liraglutide, a leading drug for the therapy of diabetes type II, have been investigated by spectroscopic measurements and MD simulations. Fluorescence measurements were used to establish the LG18 *cac* by the pyrene assay and the average dimension of LG18 aggregates by fluorescence anisotropy. DLS measurements in the concentration range between 0.47 and 36 μM have also been carried out to determine the size distribution of LG18 aggregates. In particular, we found a *cac* = 4.3 μM and a hydrodynamic radius of around 15 Å (*V_h_* ~13 nm^3^) in the concentration range between 21 and 36 μM. Interestingly, NMR measurements carried out in the 50 to 250 μM concentration range provided a hydrodynamic radius of 28 Å (*V_h_* = 92 nm^3^), suggesting the formation of larger nanostructures at these higher concentrations [21].

Fluorescence experiments and MD simulations highlighted the formation of small LG18 clusters that trigger large scale aggregation. At micromolar concentrations, DLS measurements confirmed the presence of lipopeptide clusters of nanometric dimensions, in agreement with fluorescence anisotropy measurements carried out under the same experimental conditions. On the other hand, DLS measurements on aged LG18 solutions revealed the formation of micrometric aggregates.

A peculiar fluorescence emission was observed between 380 and 460 nm under aggregation conditions, i.e., after a relatively long time since the preparation of novel LG18 solutions. This emission has been detected in the case of fibrillization of amyloid peptides, and it was ascribed to the formation of large β-sheet structures, characterized by an extended network of H-bonded peptides [23,24]. The lag time preceding the formation of mesoscopic peptide structures, randomly lasting from one week to one month, was found to critically depend on the lipopeptide concentration.

Despite the successful therapeutic activity of Liraglutide, its pharmacokinetic properties are still under scrutiny and optimization. The functionalization of the peptide scaffold at K26 with a C18 lipid chain clearly improved the LG18 capacity to permeabilize the cell membrane; it also deeply affected its aggregation properties in terms of its *cac* conditions, aggregation kinetics, and morphology.

From this point of view, it is interesting to compare the aggregation properties and binding affinity to serum proteins of Liraglutide, a once daily administration drug, and Semaglutide, a one weekly administration drug. The former shows a *cac* lower than the latter, while Semaglutide shows a markedly higher affinity for albumin [21]. This finding indicates that very long-lasting pharmacokinetics can be achieved by the accurate balance of aggregation properties dictated by the drug solubility in the administration environment and the plasma protein binding affinity.

The fine tuning of drug circulation time is finally determined by the extent of the protective action against degradation and enzymatic attack exerted by the aggregate and the formation of drug–protein conjugates.

Aggregation studies appear, therefore, to be necessary to understand the mechanism of action of therapeutic peptides and their behavior in a real environment [27]. In this regard, optical spectroscopy techniques, and in particular fluorescence spectroscopy and DLS measurements, when combined with complementary MD simulations, are invaluable to obtain information at sub-micromolar concentrations on the mechanism and time evolution of the peptide self-assembly process.

## Figures and Tables

**Figure 1 molecules-28-07536-f001:**
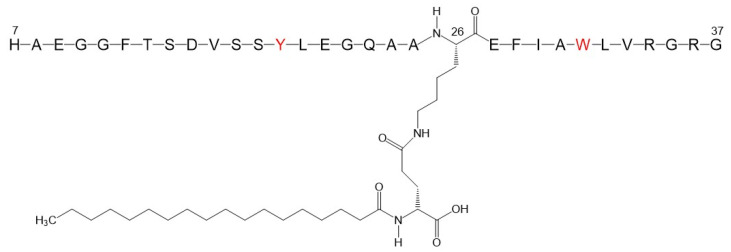
Molecular formula of the Liraglutide analogue investigated (LG18). The lipid chain, functionalizing the K26 residue, comprises 18 C atoms. The Y and W fluorescent residues are shown in red.

**Figure 2 molecules-28-07536-f002:**
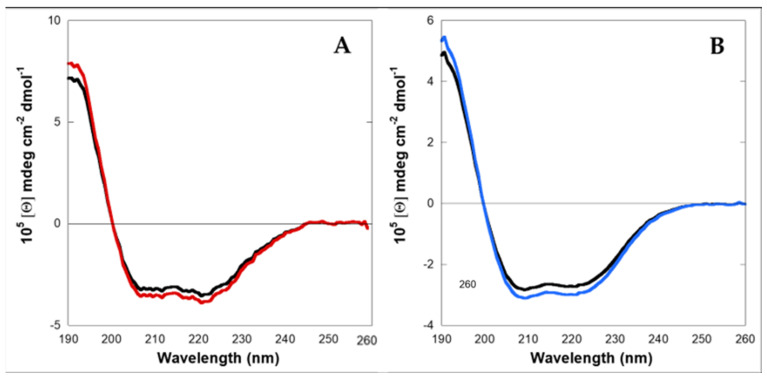
Circular dichroism spectra of LG18. (**A**) 2 μM; red: just prepared; black: after two weeks; (**B**) 10 μM; blue: just prepared; black: after two weeks.

**Figure 3 molecules-28-07536-f003:**
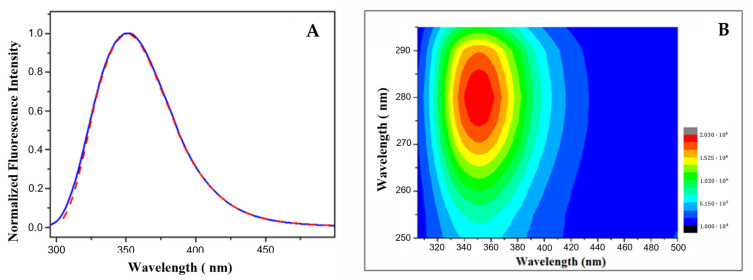
(**A**) Emission spectra of LG18 at λ_ex_ = 280 (red line) and 295 (blue line) nm; (**B**) two-dimensional excitation/emission spectrum of LG18 (PBS, pH 8.1).

**Figure 4 molecules-28-07536-f004:**
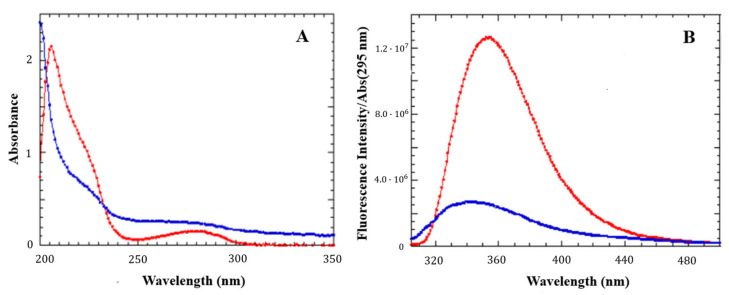
UV-vis absorption (**A**) and emission (**B**) spectra of LG18 (21 μM, pH = 8.1) at different times. Red: freshly prepared solution. Blue: after two weeks. The emission spectra were normalized by the absorbance at the excitation wavelength (295 nm).

**Figure 5 molecules-28-07536-f005:**
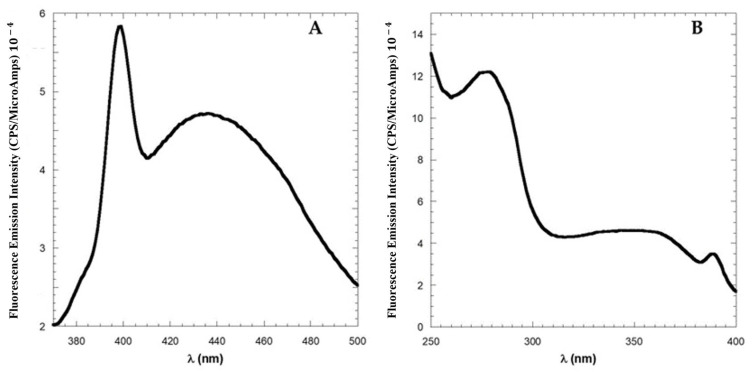
Fluorescence spectra of one month aged solutions of 10 μM LG18. (**A**) Emission spectrum (λ_ex_ = 350 nm). The sharp peak at 395 nm is the Raman band of the solvent when excited at 350 nm. (**B**) Fluorescence excitation spectrum (λ_em_ = 450 nm).

**Figure 6 molecules-28-07536-f006:**
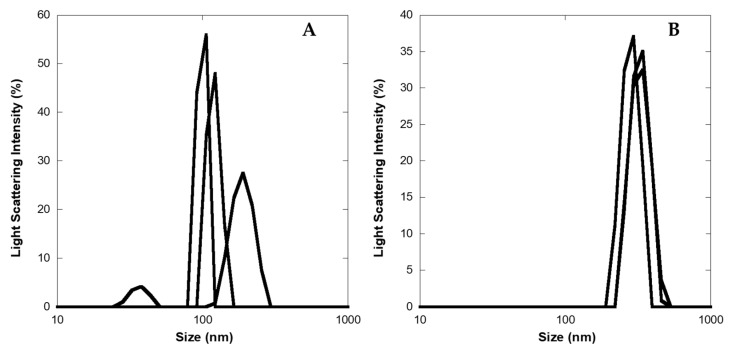
Diameter size distribution (nm) of LG18 aggregates as obtained by DLS experiments on LG18 aged (one month) solutions. (**A**) 2 μM; (**B**) 10 μM.

**Figure 7 molecules-28-07536-f007:**
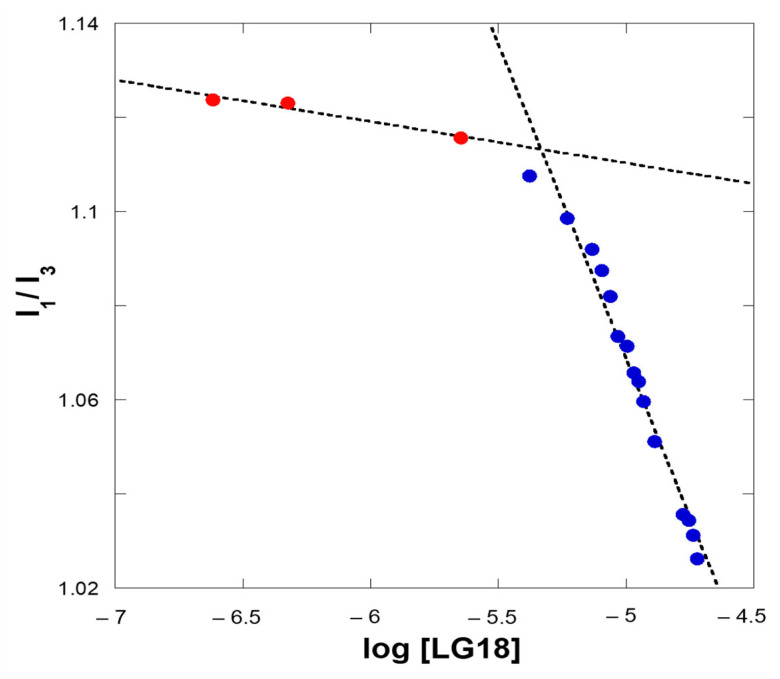
Ratio of the first and third vibronic components, i.e., I_1_/I_3_, of the 360–420 pyrene emission band as a function of micromolar LG18 concentrations (semi-log plot). The red and blue dots refer to the transition from a polar to an apolar environment, respectively.

**Figure 8 molecules-28-07536-f008:**
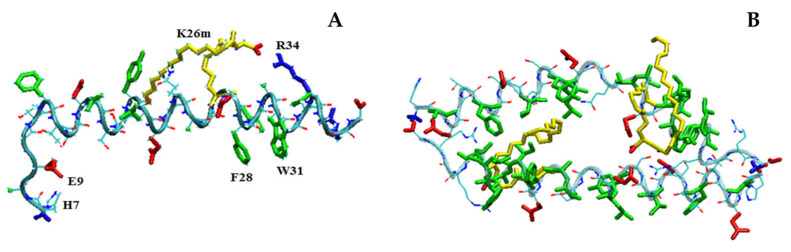
Structure of the LG18 monomer (**A**) and dimer (**B**) after 100 ns. Hydrophobic, negatively charged, and positively charged residues are shown in green, red, and blue colors, respectively. The K26m side chain is yellow.

**Figure 9 molecules-28-07536-f009:**
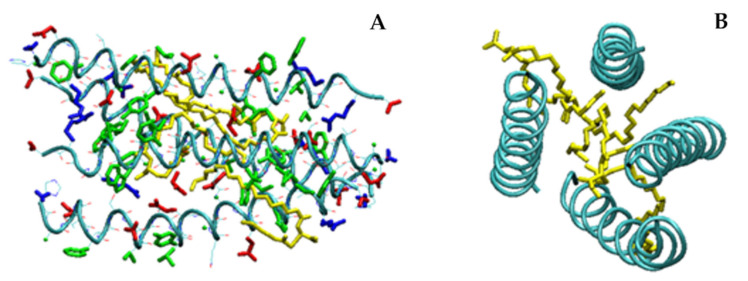
(**A**) Side and (**B**) top views of the aggregate formed by four LG18 units after 100 ns. Hydrophobic, negatively charged, and positively charged residues are shown in green, red, and blue colors, respectively. The K26m side chain is yellow. For the sake of clarity, on B only the backbones and the K26m side chains are shown.

**Table 1 molecules-28-07536-t001:** Average time decay (<τ>), fluorescence anisotropy coefficient (*r*), hydrodynamic volume (V_h_), and radius (R_h_) of LG18 micromolar solutions (λ_ex_ = 295 nm and λ_em_ = 350 nm).

Concentration (μM)	<τ> (ns)	*r*	V_h_ (nm^3^)	R_h_ (Å)
3	2.80	0.109 ± 0.001	9.3	13.0
21	3.00	0.112 ± 0.001	10.5	13.5
36	3.01	0.127 ± 0.001	13.3	14.7

**Table 2 molecules-28-07536-t002:** Gyration radii (R_g_), gyration volume (V_g_), hydrodynamic radius (R_h_), and hydrodynamic volumes (V_h_) of LG18 monomer, dimer, tetramer, octamer, and hexadecamer.

LG18	R_g_ (Å)	V_g_ (nm^3^)	R_h_, (Å)	V_h_ (nm^3^)
Monomer	13.9 ± 0.5	11.3 ± 0.6	10.8 ± 0.2	5.3 ± 0.3
Dimer	25.1 ± 1.1	66.2 ± 1.2	10.9 ± 0.2	5.4 ± 0.3
Tetramer	36.1 ± 0.7	197.1 ± 0.8	11.2 ± 0.1	5.9 ± 0.2
Octamer	38.5 ± 0.5	239.0 ± 0.6	12.1 ± 0.1	7.4 ± 0.2
Hexadecamer	45.2 ± 0.5	386.8 ± 0.6	13.8 ± 0.1	11.0 ± 0.2

**Table 3 molecules-28-07536-t003:** Number of LG18 and H_2_O molecules and size of the cubic simulation box used in the MD simulations.

No. of LG18 Molecules	No. of H_2_O Molecules	Dimensions of the Simulation Box (nm^3^)
1	11,068	7.00 × 7.00 × 7.00
2	10,934	7.01 × 7.01 × 7.01
4	16,319	8.02 × 8.02 × 8.02
8	26,931	9.55 × 9.55 × 9.55
16	47,434	11.53 × 11.53 × 11.53

## Data Availability

Data are contained within the article and supplementary materials.

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
