# Peer review of "A Spectroscopic and Molecular Dynamics Study on the Aggregation Properties of a Lipopeptide Analogue of Liraglutide, a Therapeutic Peptide against Diabetes Type 2"

_molecules, 2023, doi:10.3390/molecules28227536_

Round 1

Reviewer 1 Report

Comments and Suggestions for Authors

In this work, the authors analyzed the aggregation properties of Liraglutide analogue LG18 via spectroscopic and molecular dynamics studies. Their findings help evaluate the aggregation behaviors of LG18 and its pharmacokinetic properties and provide molecular insights on understanding the mechanism of action of therapeutic LG18 and other peptides. I would recommend a major revision for this manuscript to publish on Molecules after addressing the issues listed below:

1.       This manuscript did not present any data on the therapeutic properties against diabetes type 2. Therefore, the term “a therapeutic peptide against diabetes type 2” seems not suitable to appear in the title.

2.       The spectroscopic results suggest Y19*-W31* transfer which only occurs when their distance is close enough. Do the MD simulation results support this finding, in the case of dimer, tetramer, or larger aggregates?

3.       Long-term aging of LG18 after one month results in the formation of peptide amyloids. Imaging clues such as TEM or cryo-EM are recommended for supporting this statement.

Other specific points:

1.       The assignment of absorbance bands in UV-Vis spectra is inaccurate. l=211 nm is attributed to n-π* transitions, while l=280 nm is attributed to π-π* transitions, respectively.

2.       The CD spectra show a higher signal intensity of LG18 at 2 mM than 10 mM. Can the authors explain why they are not concentration-dependent?

3.       Figure 6 was not shown in the text.

4.       In the experimental section, the parameters a, b, and c of the cubic simulation box are suggested to show rather than the total box volume. 

Reviewer 2 Report

Comments and Suggestions for Authors

The author performed spectroscopy methods and molecular dynamics simulation to investigate the aggregation properties of an LG analogue, denoted in the following as LG18.

The paper's English is generally correct, but there are some raised questions, comments, and suggestions that the author should address before final publication:

(1) For Figure 2 of lines 98-99, the size for labels and numerical coordinates for the x- and y-axes are too small to read. Please try to keep all labels clearly visible and all figures consistent.

(2) For Figure 3 of lines 117-119, 4 page, (a) labels and numerical coordinates are in bold, which is inconsistent with other figures. Please be consistent throughout the text. (b) The caption of the Figure 3 contains A and B in lines 118-119, 4 pages, but there are no marks A or B on the Figure 3.

(3) For Figure 5, why is the x-axis label suddenly placed on the right side in line 167? In Figure 7 in line 235, the x- and y-axis labels are oddly positioned.

(4) Markers A and B are located in the upper left corner of Figure 6 and 8 in line 209 and line 254. The markings in Figures 2, 4, 5 and 9 (line 98, 149, 167, 270) are on the upper right. In an article, the size of the markup varies from large to small, which really looks very inappropriate and unprofessional for a scientific paper. Especially, the marks of Figure 9 are extremely big.

(5) For Table 2 in lines 293-294, the width is not suitable. Please keep consistent with Table 1.

(6) The types of references should be consistent. For example, reference 13 is from the Journal of Molecular and Cellular Endocrinology. The full name of the journal is written, while most other journals are mostly abbreviated.

(7) There are places in the text where redundant blank space appears multiple times. For example, line 342, please correct it carefully.

(8) The representation used in line 280 is Figure S7, and Figure * is used in many other places in the text. Please be consistent throughout the text.

(9) Please maintain the quality of the graphics throughout the manuscript and support information, they are somewhat distorted.

Comments on the Quality of English Language

The paper's English is generally correct.

Round 2

Reviewer 1 Report

Comments and Suggestions for Authors

satisfied with the improvement. recommend acceptance in the present form

Reviewer 2 Report

Comments and Suggestions for Authors

It appears that Figure B in Figure 3 is blurry, and the numbers on the colorbar are not legible. It is better to modify it.